# Study on the corrosion behavior and mechanical response of weakly cemented sandstone in alkaline solutions

**Jie Zhang**[1], **Qingsong Zhuo**[1]*, **Qian Zheng**[2], **Bin Wang**[1], **Mingang Zhang**[3], **Xiaoyu Zhao**[3], **Jigang Geng**[3], **Xiaoshi Li**[3], **Ruoyu Bao**[4]

**1** College of Energy, Xi'an University of Science and Technology, Xi'an, China, **2** College of Intelligent Manufacturing and Information Engineering, Shaanxi Energy Institute, Xianyang, China, **3** Department of Medical Imaging, Xi'an Daxing Hospital, Xi'an, China, **4** Information Institute of the Ministry of Emergency Management of the PRC, Beijing, China

* zhuo_simon@163.com

**Data Availability Statement:** All relevant data are within the manuscript and its Supporting information files.

## Abstract

This study examines the corrosion characteristics of weakly cemented sandstone under alkaline conditions, evaluating the effects of varying pH levels on its macroscopic degradation, micro-porosity, and mechanical properties, notably uniaxial compressive strength. Findings reveal that heightened alkalinity exacerbates rock damage, although a temporary alleviation in mass loss occurs between pH 9 and 11 due to pore clogging by complexes formed from cations like $Ca^{2+}$ and $Mg^{2+}$. Increased alkalinity induces marked changes in pore features, with an observed rise in pore numbers, transformation of pore shapes from elongated to more spherical, and adjustments in porosity, pore size, and roundness. Furthermore, the study confirms a decline in both the rock's compressive strength and elastic modulus as pH rises. These revelations shed light on the role of pH in the corrosion behavior of weakly cemented sandstone under alkaline conditions, providing a fresh perspective for understanding its corrosion mechanisms in such environments.

## Introduction

Rocks, as major components of the Earth's crust, have a decisive impact on the stability of geotechnical engineering due to their physical and mechanical properties. Although most rocks exhibit high strength and compactness, their internal microstructures are particularly sensitive to chemical corrosion, largely due to natural defects and microfractures, potentially weakening their macroscopic mechanical capabilities [1–3]. The mechanisms of rock damage under the influence of chemical solutions is a broadly concerned issue, encompassing theories such as the change in pH values at the tips of rock cracks proposed by Wiederhorn [4], analyses by Feucht and others on the impact on the mechanical strength of sandstone [5], research by M.G. Karfakis and colleagues using fracture mechanics to investigate the intrinsic mechanisms of changes in rock properties [6], and studies by T. Critelli and others on the dissolution rates of rock bodies [7, 8]. Wang and colleagues utilized two-dimensional

**Funding:** The author(s) received no specific funding for this work.

**Competing interests:** The authors have declared that no competing interests exist.

(2D) micro-computed tomography combined with three-dimensional (3D) volume reconstruction methodologies to scrutinize the dispersion of pores and fractures within sandstones from a microscopic viewpoint, which in turn guides the understanding of macroscopic characteristics [9]. The research on the deformation and failure mechanisms of rocks under the coupled action of chemical solutions has also attracted attention from scholars both domestically and internationally, including Feng Xiaoting and Ding Wuxiu [10–12], and the rock damage models established by Chen Sili, Qiao Liping, and others [13–15]. Tan Xianjun et al. analyzed the deterioration phenomena of rock mass by examining parameters such as strength, deformation characteristics, elastic modulus, and cohesive strength through uniaxial and triaxial experiments [16], while Chen Wei and others have analyzed from macro to micro the corrosion mechanisms of acidic groundwater on coal and rock bodies [17]. Yang Xiurong and others have used SEM to analyze the damage mechanisms of the micro-structure of sandstone [18, 19], and Fang Xinyu and others have studied the mechanical characteristics of yellow sandstone under chemical corrosion and freeze-thaw conditions [20–22]. Li Xinping introduced a model for the evolution of sandstone damage [23–25], and Hao Li and others proposed and validated a damage model for rock bodies under cyclic loading [26]. Lin Yun and others established a statistical damage constitutive model describing the damage evolution pattern of sandstone [27, 28]. In studies exploring the impact of chemical corrosion on the dynamic fracture mechanics of limestone, Yu Liyuan and others found significant effects on the microstructure and mineral elements of rock bodies [2, 29, 30]. Su Xuexue and colleagues investigated the influence and micro-mechanisms of pH on the disintegration of red-layer mudstone on the slopes of the Three Gorges Reservoir [31], and Huang Ming and others developed theoretical models for the disintegration experiments of shale under various acidic and alkaline solutions [32]. Feng Xuezhi and others focused on the impact of chemical solution corrosion and freeze-thaw cycles on rock body damage mechanisms [33]. In the mining areas of Western China, weakly cemented rock bodies such as mudstone, sandstone, and interbedded mud-sandstone may undergo significant changes under the action of water [34, 35], especially in areas with unique hydrogeological conditions where groundwater is predominantly alkaline, leading to a reduction in rock strength [36]. Global research has intensively investigated the mechanisms of rock degradation induced by chemical corrosion, with a particular focus on its pivotal role in geotechnical stability. These inquiries span theories on pH fluctuations at fracture tips, the impact of chemical corrosion on sandstone strength, the evolution of rock properties from a fracture mechanics perspective, and the rates of rock dissolution, among other aspects. The application of advanced imaging techniques has refined our understanding of rock macro-behavior, while numerous scholars have made significant contributions to elucidating the deformation and failure mechanisms of rocks under chemical attack, the pH-dependent variations in mechanical properties, micro-corrosion processes, and theoretical models of rock erosion in varied media. Despite extensive studies conducted in major engineering projects and exceptional geological settings, research specifically addressing the mechanical properties of weakly cemented sandstone in alkaline environments, as in the Shaanbei mining region, remains scarce. Therefore, this paper, building upon previous research, systematically conducts mechanical property tests on weakly cemented sandstone in the Shaanbei mining area under alkaline water corrosion. The objective is to thoroughly analyze the impact of alkaline solutions on the mechanical properties of rocks and investigate their corrosion mechanisms, providing theoretical and experimental support for the development of rock mechanics constitutive models.

## Experimental materials, equipment, and methods

### Rock specimens and experimental equipment

The weakly cemented sandstone samples required for this experiment were collected from the roof rock layer of the 30208 belt conveyor roadway in Longhua Coal Mine, Sunjiacha Town, Yushenfu mining area in Northern Shaanxi. The cores were taken from the roof according to the distribution and structural characteristics of the coal seams in the mine field, and all selected cores were from the same location within the roadway. After meticulous classification and identification, the cores were processed in the laboratory into standard cylindrical specimens with diameters of 50 mm and heights of 100 mm (for compression), 25 mm (for tension), and 50 mm (for shear) according to the national "Standard for Test Methods of Rock" (GB/T 50218–2014). All specimens were precision machined to ensure that their end face perpendicularity, parallelism, and flatness met the precision required by national standards. After removing any visible defects, the average density of the specimens in their natural water-containing state was 2.66 g/cm$^3$. Mechanical property testing was completed at the MTS laboratory of Xi'an University of Science and Technology, using an HCT series microcomputer controlled electro-hydraulic servo universal testing machine. This machine is equipped with an advanced electro-hydraulic servo control system, which can achieve closed-loop control of load and displacement to ensure high precision of the experiments. During the immersion of rock samples in chemical solutions, the pH was measured using a high-precision pH meter provided by Lichen Technology, with a highly sensitive probe and three-point calibration function ensuring measurement accuracy to 0.01. The mass change of the rock specimens was monitored by a Mengchuang electronic scale with an accuracy of 0.1 g. SEM scanning analysis was conducted using an Axio Scope. A1 polarized light microscope to investigate the microstructural characteristics of weakly cemented sandstone under various alkaline immersion conditions. The operational parameters of the microscope comprised a principal voltage of 100V, power consumption of 140W, with specimen dimensions required to be at least 30mm×60mm×90mm. Additionally, a thin section slicing thickness of 0.03mm was adopted to guarantee clear observation and meticulous analysis of the intricate structures. Concurrently, through the application of pseudocolor enhancement technology, the visual quality of SEM images was improved, enabling a more intuitive analysis and description of the microscale damage and corrosion in the rock mass. Fig 1 displays the processing and testing equipment for the rock specimens.

### Preparation of chemical solutions

To verify the types of groundwater in the mining areas of Northern Shaanxi, this study selected the seepage water from the roof of the $3^{-1}$ coal seam in Longhua Mine, Shenmu City for chemical composition analysis. Field measurements indicated that the average annual temperature in the $3^{-1}$ coal seam roadway is approximately 20.8˚C, with relative humidity ranging from 93% to 95%. When collecting water samples, 500 mL plastic bottles, rinsed three to four times with the raw water and corresponding samples, were used. The pH value of the water samples was determined using an onsite water quality monitor and preserved under sealed conditions at temperatures between 0 to 4˚C. Sample filtration was conducted using a microporous filter membrane with a diameter of 0.45μm for cation testing. The samples were treated with dilute nitric acid in the laboratory, and the concentrations of anions ($SO_4^{2-}$, $Cl^-$) were determined by ion chromatography, while the concentration of $HCO_3^-$ was measured by titration; cation concentrations ($Ca^{2+}$, $Na^+$, $Mg^{2+}$) were tested using an inductively coupled plasma emission spectrometer. To ensure the accuracy of the tests, an ion conservation analysis was conducted. If

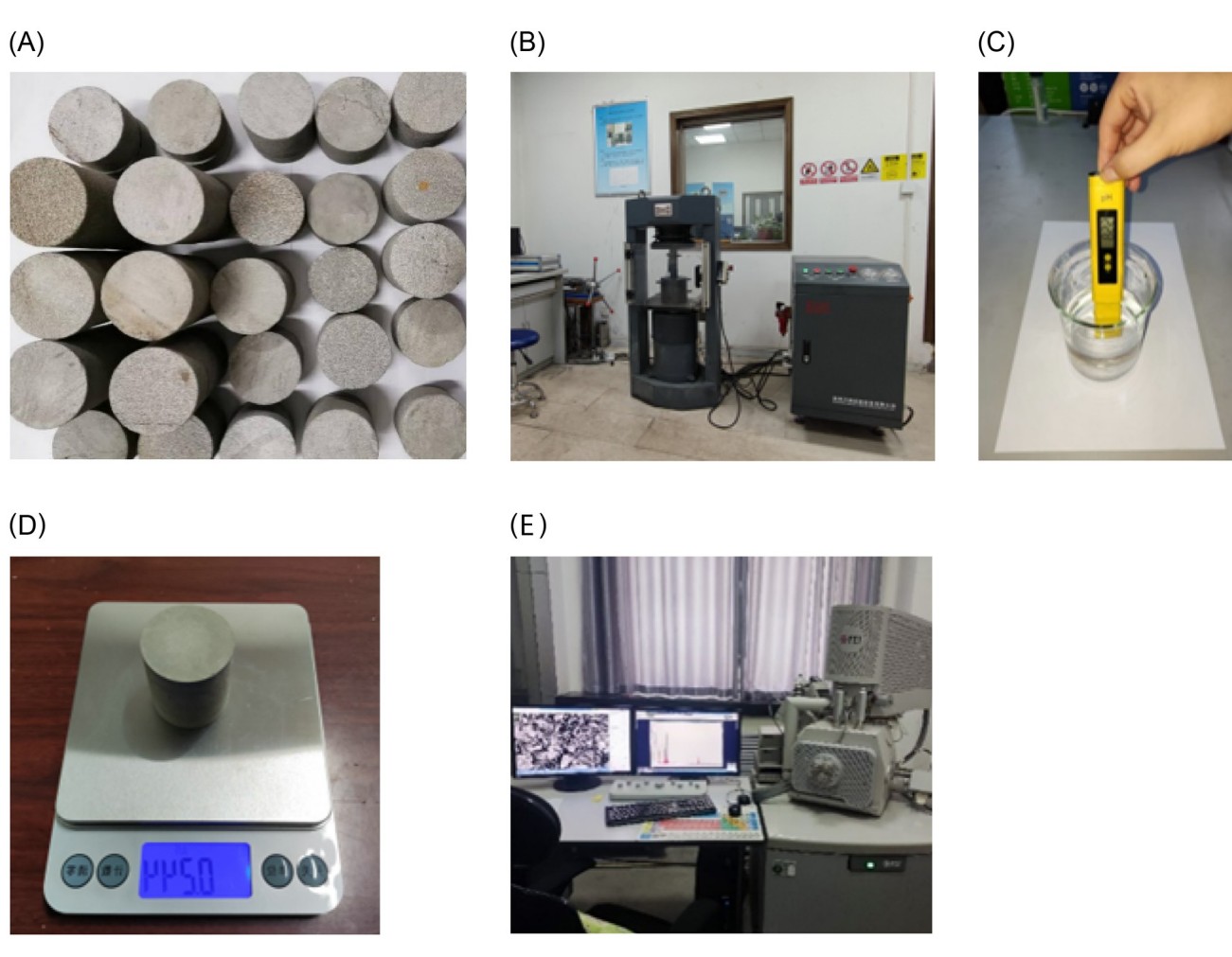

**Fig 1. Rock specimens and testing equipment.** (A) Processed specimen. (B) Pressure testing machine. (C) pH testing pen. (D) Electronic scale. (E) Polarized light microscope.

the ratio of ion concentration was within a 5% range, it was considered conserved; otherwise, the measurement was retaken. Table 1 is a statistical table of the groundwater chemical characteristics for the $3^{-1}$ coal seam at the Longhua Mine. The groundwater is slightly alkaline, with an average pH value of 8.75 for the $3^{-1}$ coal seam. The groundwater exhibited a slightly alkaline nature, with an average pH value of 8.75 for the $3^{-1}$ coal seam. The ion concentration test results indicated a high degree of mineralization, primarily characterized by high contents of $Ca^{2+}$, $Na^+$, $Mg^{2+}$, $Cl^-$, $SO_4^{2-}$, and $HCO_3^-$, with the water chemistry type being predominantly $HCO_3^-$-Ca-Na type.

To simulate the impact of the alkaline hydrochemical environment of shallow coal seams on the mechanical properties of surrounding rocks, solutions with varying ion concentrations and pH levels were prepared, taking into consideration the primary cations ($K^+$, $Na^+$, $Ca^{2+}$, $Mg^{2+}$) and anions ($Cl^-$, $SO_4^{2-}$, $HCO_3^-$) present in the percolating water of the $3^{-1}$ coal seam roof. The solution preparation scheme for the water chemistry is shown in Table 2. Solutions with pH values of 7, 9, and 11 were prepared by adding corresponding salts, including anhydrous calcium chloride, sodium chloride, magnesium chloride, and potassium sulfate. These solutions were used to immerse the surrounding rock samples, aiming to observe changes in rock

**Table 1. Chemical characteristics of groundwater in the Longhua Mine's $3^{-1}$ coal seam.**

| pH value and ion concentration | | $3^{-1}$ coal seam |
|---|---|---|
| pH | | 8.75 |
| $K^+$ | mg/l | 61.01 |
| $Na^+$ | mg/l | 416.92 |
| $Ca^{2+}$ | mg/l | 101.18 |
| $Mg^{2+}$ | mg/l | 93.79 |
| $SO_4^{2-}$ | mg/l | 590.64 |
| $Cl^-$ | mg/l | 1728 |
| $HCO_3^-$ | mg/l | 673.39 |
| $NO_3^-$ | mg/l | 66.04 |
| $NO_2^-$ | mg/l | 20.21 |

mechanical properties under different alkaline conditions. Deionized distilled water was prepared using the advanced ultrapure water production system at Xi'an University of Science and Technology's laboratory. The equipment used for immersing the samples was a DZF-1 vacuum drying oven. Fig 2 illustrates the drying process of the immersed samples.

## Experimental method

Regarding the experimental procedure, rock samples were placed in open PPS acid-base immersion tanks to ensure continuous contact between the solution and the external environment. Air conditioning was employed to control the indoor temperature. During the immersion process, regular records of the mass variation ($\Delta m$) of the samples were maintained, and data changes were monitored. The pH value of the immersion solution was measured using a pH meter, and the trend of pH changes was recorded. After immersion, the rock samples underwent drying and polishing treatments to ensure that the end faces and side surfaces met testing standards and maintained a smooth and vertical appearance. Prior to conducting uniaxial compression tests on the rock samples, a uniform application of Vaseline was applied to both the upper and lower ends of the samples to minimize friction's influence on the test results. The testing employed a displacement loading method, with the upper loading device kept stable, and a lower loading rate of 0.06 millimeters per minute was applied. Throughout this process, the axial load and axial displacement of the rock sample were monitored in real-time, and deformation and failure processes were recorded. Data changes and numerical variations were also concurrently documented and analyzed.

## Analysis of chemical damage mechanisms under alkaline corrosion

Weakly cemented sandstone primarily consists of a framework composed of quartz and feldspar, with pore-filling materials dominated by clay minerals. These two components collectively determine its structural characteristics. Quartz is chemically composed of $SiO_2$, and the

**Table 2. Preparation scheme of water chemical solutions.**

| Solution type | Solution components | Solution pH value | Solution concentration (mg/l) |
|---|---|---|---|
| Hydrogen ion concentration of 7 solution | $K_2SO_4$, $CaCl_2$, $NaCl$, $MgCl_2$ | 7 | 0.1 |
| Hydrogen ion concentration of 9 solution | $K_2SO_4$, $CaCl_2$, $NaCl$, $MgCl_2$ | 9 | 0.1 |
| Hydrogen ion concentration of 11 solution | $K_2SO_4$, $CaCl_2$, $NaCl$, $MgCl_2$ | 11 | 0.1 |

(A)

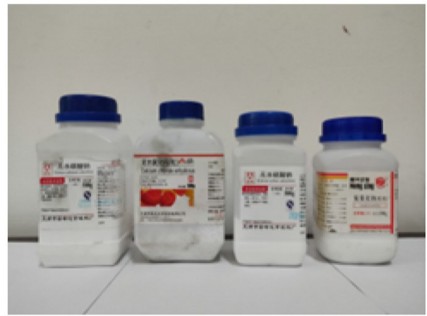

(B)

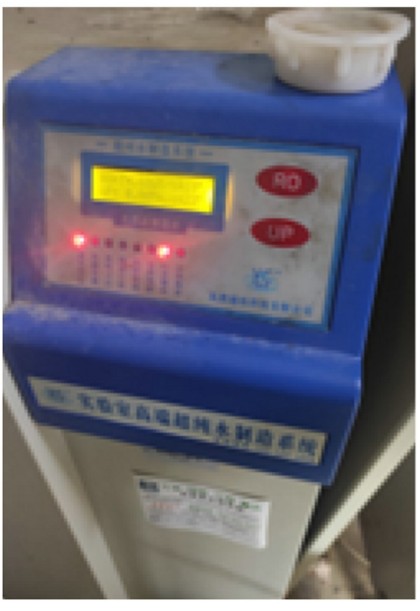

(C)

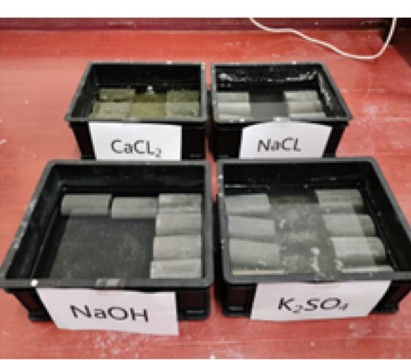

(D)

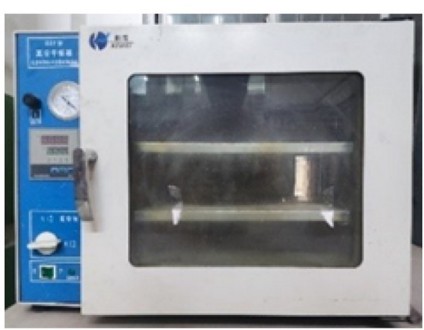

**Fig 2. Soaking materials and drying equipment.** (A) Ion concentration preparation chemicals. (B) Ultrapure water manufacturing equipment. (C) Soaked rock specimens. (D) DZF vacuum drying oven.

chemical formula for potassium feldspar is $KAlSi_3O_8$. Clay minerals include kaolinite $(Al_2[Si_2O_5](OH)_4)$, montmorillonite $(Na_{0.33}(Al,Mg)_2Si_4O_{10}(OH)_2)$, and others. As the solution's alkalinity increases, changes in these mineral components indicate their involvement in hydrochemical reactions. The specific chemical reactions are as follows:

1. In a neutral solution with a pH value of 7, the framework and pore-filling materials of the rock primarily undergo hydrolysis reactions, slowly releasing trace amounts of metal ions such as aluminum and magnesium. These ions subsequently form aluminate ions in the solution. This process may result in minor mineral alteration, although the basic structure of the rock is preserved, long-term exposure may lead to slight changes in its physical and chemical properties.

$$KAlSi_3O_8 + H^+ + 4H_2O \rightarrow K^+ + Al^{3+} + 3H_4SiO_4 \qquad (1)$$

2. When the pH value is raised to 9, the release rate of aluminum, magnesium, and other ions from potassium feldspar and clay minerals accelerates, particularly that of magnesium ions. This accelerated release may lead to partial loosening of mineral structures. The dissolution of aluminum and the subsequent formation of aluminum hydroxide ions may attach to the surface of clay minerals, disrupting the lattice structure of potassium feldspar and reducing its physical mechanical strength. In this environment, kaolinite and montmorillonite may transform into silicates and other new compounds, as shown in Eq (3).

$$KAlSi_3O_8 + 8H_2O \rightarrow K^+ + Al(OH)_4^- + 3H_4SiO_4 \tag{2}$$

$$NaAlSi_3O_8 + 8H_2O \rightarrow Na^+ + Al(OH)_4^- + 3H_4SiO_4 \tag{3}$$

3. In a strongly alkaline solution with a pH value of 11, the stability of the rock mass is severely threatened. A large and rapid release of potassium, magnesium, and aluminum ions leads to rapid mineral degradation. Under strong alkaline conditions, kaolinite first undergoes dehydroxylation reactions, while $SiO_2$ in the strongly alkaline environment undergoes hydrolysis to produce a large amount of silicate ions ($SiO_3^{2-}$). Aluminum also forms aluminum salts under alkaline conditions. Montmorillonite, a 2:1 layered silicate mineral with a negative charge, undergoes ion exchange with cations such as calcium ions in alkaline aqueous solutions. Along with the dissolution of silicon, aluminum, and magnesium, this results in the formation of corresponding silicate ions, aluminum salts, and magnesium salts. These reactions severely disrupt the framework structure of the rock, leading to a significant reduction in the structural performance of weakly cemented sandstone. At the micro and meso levels, these changes manifest as the development of pores and cracks, while at the macro level, they result in damage to the mechanical properties of the rock.

$$KAl_3Si_3O_{10}(OH)_2 + 2OH^- + 10H_2O \rightarrow K^+ + 3Al(OH)_4^- + 3H_4SiO_4 \tag{4}$$

$$Al_2Si_2O_4(OH)_4 + 4OH^- \rightarrow 2Al(OH)_4^- + 2SiO_3^{2-} + 2H_2O \tag{5}$$

$$Na_{0.33}(Al, Mg)_2Si_4O_{10}(OH)_2 + 6OH^- \rightarrow \\ 2Al(OH)_4^- + 2SiO_3^{2-} + 2Mg(OH)_4^{2-} + 0.3Na^+ \tag{6}$$

## Results analysis

### Analysis of mass damage to rocks in alkaline solutions

Immersing specimens in alkaline solutions inevitably leads to chemical reactions that cause mineral components to leach out, affecting the mass of the rock before and after corrosion. The extent of mass change varies among specimens with different degrees of corrosion. Prior to measurement, surface moisture is removed until no obvious droplets remain, followed by drying and weighing on an electronic scale. The relative mass change ($\Delta m$) of a sample in various hydrated chemical solutions is recorded over time, revealing the process of water-rock chemical action. Table 3 lists the mass values of the samples before and after stabilization of the chemical action, while Fig 3 shows the curve of mass damage to rock samples in different pH solutions over time.

**Table 3. Mass loss $\Delta$m (g) of corrosion specimens in different pH solutions.**

| Solution pH | Corrosion time (days) | | | | | | | | |
|---|---|---|---|---|---|---|---|---|---|
| | Sample No | 1d | 7d | 14d | 21d | 28d | 35d | 42d | 49d |
| pH = 7 | 1 | 0.72 | 0.97 | 0.91 | 0.92 | 0.96 | 0.98 | 0.98 | 0.97 |
| | 2 | 0.61 | 0.86 | 0.86 | 0.86 | 0.87 | 0.91 | 0.88 | 0.87 |
| | 3 | 0.49 | 0.82 | 0.81 | 0.81 | 0.83 | 0.83 | 0.84 | 0.83 |
| Average | | 0.60 | 0.87 | 0.86 | 0.88 | 0.89 | 0.90 | 0.90 | 0.89 |
| pH = 9 | 1 | 0.88 | 1.60 | 1.62 | 1.62 | 1.63 | 1.63 | 1.64 | 1.63 |
| | 2 | 0.85 | 1.51 | 1.56 | 1.55 | 1.59 | 1.57 | 1.57 | 1.57 |
| | 3 | 0.81 | 1.49 | 1.51 | 1.53 | 1.49 | 1.51 | 1.52 | 1.51 |
| Average | | 0.85 | 1.53 | 1.56 | 1.57 | 1.57 | 1.57 | 1.58 | 1.57 |
| pH = 11 | 1 | 1.39 | 1.69 | 1.71 | 1.70 | 1.69 | 1.67 | 1.66 | 1.66 |
| | 2 | 1.33 | 1.60 | 1.63 | 1.63 | 1.61 | 1.59 | 1.60 | 1.61 |
| | 3 | 1.29 | 1.59 | 1.59 | 1.57 | 1.57 | 1.58 | 1.56 | 1.54 |
| Average | | 1.34 | 1.60 | 1.64 | 1.63 | 1.62 | 1.61 | 1.60 | 1.60 |

Trend analysis from Fig 3 indicates: (1) With increasing corrosion time, mass loss in all alkaline solutions tends to increase, especially significant in strong and medium alkaline solutions, while the impact of neutral solution is relatively minimal. (2) Initially, the mass change ($\Delta m$) is substantial, stabilizing after 6 days, indicating intense water-chemical reactions initially, weakening later. (3) After 50 days of corrosion, the impact of different pH solutions on rock mass is distinct, with the average mass loss in pH = 11 solution being the greatest at 1.60g, equating to a daily loss rate of $3.2 \times 10^{-2}$g; loss in pH = 9 solution is slightly lower, and the rate in pH = 7 solution is the lowest.

For varying corrosion times, fitting the mass loss $\Delta m$ of specimens in different pH chemical solutions yields the curve as shown in Fig 3. The formulae derived for mass loss $\Delta m$ over time $x$ in various corrosion states are:

$$\Delta m = m_0 + A_1(1 - \text{epx}^{x/t_1}) + A_2(1 - \text{epx}^{x/t_2}) \tag{7}$$

Eqs (7), (8), (9) and (10) detail these relationships.

$$\Delta m = 2.796 + 1.25(1 - \text{epx}^{x/0.04}) + 0.37(1 - \text{epx}^{-x/2.70})\ R^2 = 0.998 \tag{8}$$

$$\Delta m = -0.016 + 0.79(1 - \exp^{-x/2.30}) + 0.79(1 - \exp^{-x/2.30})\ R^2 = 0.981 \tag{9}$$

$$\Delta m = -0.002 + 0.09(1 - \exp^{-x/25.15}) + 0.84(1 - \exp^{-x/1.52})\ R^2 = 0.995 \tag{10}$$

Through fitting analysis of the relationship between mass loss $\Delta m$ in different pH solutions and corrosion time, we find these variations can be approximately described using a Boltzmann function, showing a high correlation between $\Delta m$ and the pH of the chemical solution. The main component of quartz is $SiO_2$, an alkaline oxide. Under the action of a pH = 7 solution, due to the presence of abundant $Ca^{2+}$ in the solution, hydrolysis reactions of calcite and calcium substances are inhibited, and clay minerals in weakly cemented sandstone adsorb $Ca^{2+}$, clogging pores and fractures, impeding migration, thereby reducing the solubility of $SiO_2$. Reactions in various pH solutions are detailed in Eqs (11) to (14).

$$2R^{n-} + nCa^{2+} \rightarrow R_2Ca_n \tag{11}$$

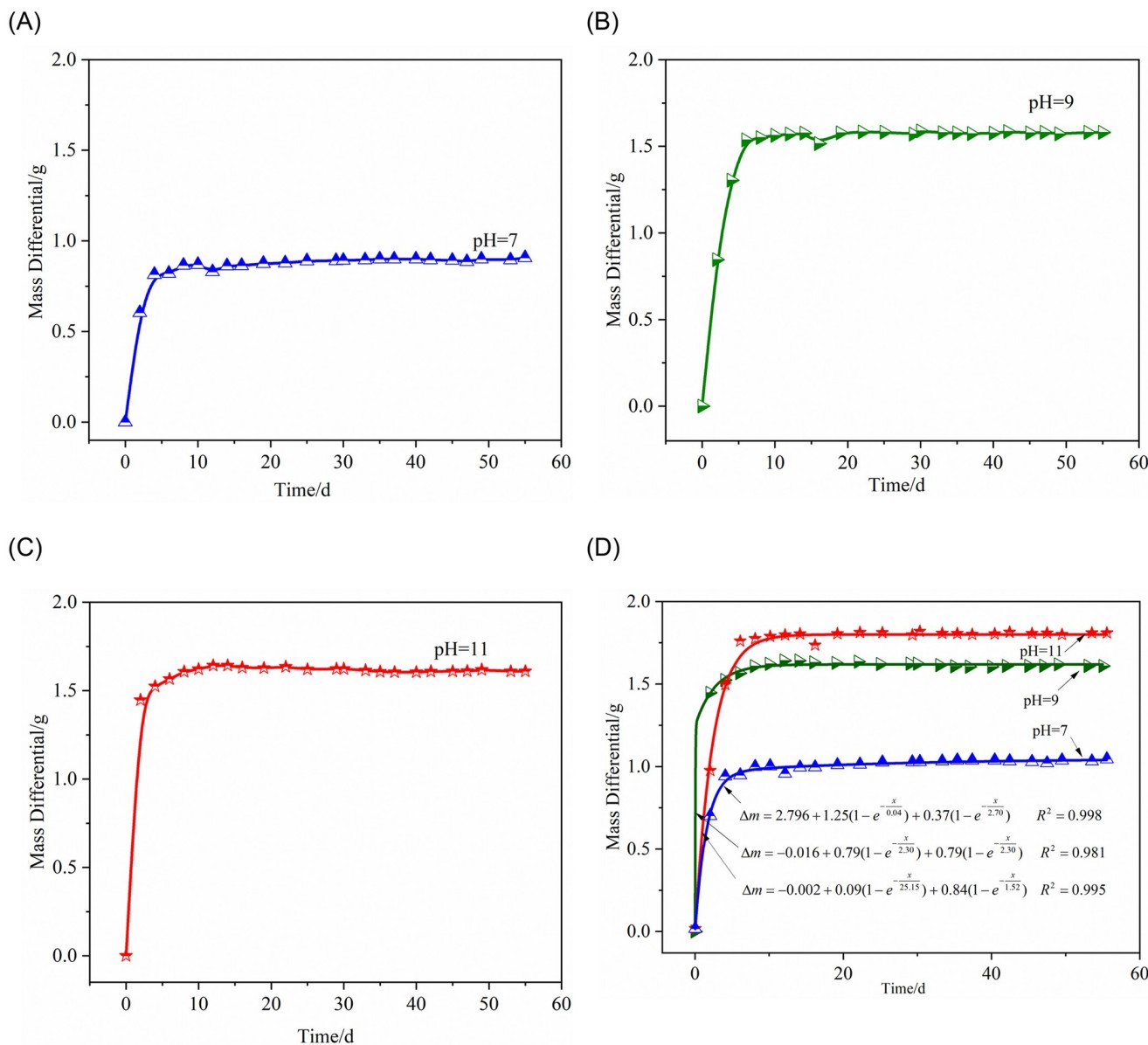

**Fig 3. Pattern of rock mass loss under hydrochemical action over corrosion time.** (A) Mass difference of pH = 11 specimen. (B) Mass difference of pH = 9 specimen. (C) Mass difference of pH = 7 specimen. (D) Fitting curve.

When the aqueous solution is at pH = 7, the solubility of $SiO_2$ is very low, and the dissolution reaction of $SiO_2$ can be written as:

$$SiO_2 + 2H_2O \rightarrow H_4SiO_4 \tag{12}$$

When the pH of the solution is $\geq 9$, $SiO_2$ undergoes hydrolysis to form $H_4SiO_4$. $H_4SiO_4$ further decomposes and can be broken down into $H_3SiO_4^-$ and $H_2SiO_4^{2-}$ through the following

reactions, which can be represented by the chemical Eq as:

$$H_4SiO_4 \rightarrow H_3SiO_4^- + H^+ \tag{13}$$

$$H_4SiO_4 \rightarrow H_3SiO_4^{2-} + H^+ \tag{14}$$

After 49 days, *X*-ray diffraction tests show significant changes in the composition of rock specimens, with quartz's mass percentage dropping from 34.4% to 28.1%, a 20% decrease. $SiO_2$ precipitation increases significantly, and the alkaline solution corrodes the rock specimens, reducing their mass. Over the same period, specimen mass $\Delta m$ shows pH = 11 > pH = 9 > pH = 7.

### Analysis of the pore development patterns in weakly cemented sandstone

In the previous section, the mass damage to rocks by different alkaline solutions was analyzed. This section builds upon the last, employing microscopic methods to analyze the distribution of pores and fractures in weakly cemented sandstone under various alkaline aqueous environments. The approach used here includes analyzing parameters such as the rock's pore size, porosity, roundness, and orientation frequency.

Porosity: This refers to the ratio of the area of all pores to the total area in SEM images, characterized by the average area of pores within each pore size range.

Pore size: Defined as the longest chord through a pore, representing the size of the pore.

Roundness: $R = P^2/4\pi A$, where $P$ is the perimeter of the pore and $A$ is its area. Roundness characterizes the shape variation and roundness of pores, representing the shape factor of pores. The larger the shape factor, the more elongated the pore.

Using AVIZO software for domain value segmentation of SEM images of weakly cemented sandstone, appropriate domain values were selected that encompass the pores on the rock surface. Small spots in the images were excluded, and the blue areas represent the distribution of pores in the weakly cemented sandstone under different alkaline water conditions, with other areas indicating undamaged rock base, as shown in Fig 4.

Table 4 provides the statistics on the micro-pore distribution of weakly cemented sandstone under various alkaline water conditions. The distribution of microscopic pores in weakly cemented sandstone under different alkaline water conditions was statistically analyzed, with the following patterns observed:

1. Under deionized distilled water soaking conditions, the rock surface exhibited extensive micropores and microcavities, with a surface porosity of 29.09%. The pore diameters ranged from 6.03e-6 to 0.0008 $\mu$m, with an average of 2.05e-5 $\mu$m and a variance of 8.16e-10. The average pore area was 2.15e-10 $\mu$m$^2$. The roundness of the pores ranged from 0.97 to 1094.45, with an average roundness of 6.74 and a variance of 663.

2. Under 0.01 mol/L $CaCl_2$ pH = 7 soaking conditions, the surface porosity increased to 31.91%, representing a 9.69% increase compared to deionized distilled water. The pore diameters ranged from 8.20e-6 to 0.0011 $\mu$m, with an average of 2.43e-5 $\mu$m and a variance of 1.25e-9, indicating uniformity in pore sizes. The average pore area was 2.55e-10 $\mu$m$^2$. The roundness of the pores ranged from 0.95 to 925.00, with an average roundness of 4.89 and a variance of 607.

3. Under 0.01 mol/L $CaCl_2$ pH = 9 soaking conditions, there was a further increase in micropores and microcavities on the rock surface. The surface porosity was 33.62%. The pore diameters ranged from 1.03e-6 to 0.0013 $\mu$m, with an average of 2.64e-5 $\mu$m and a variance

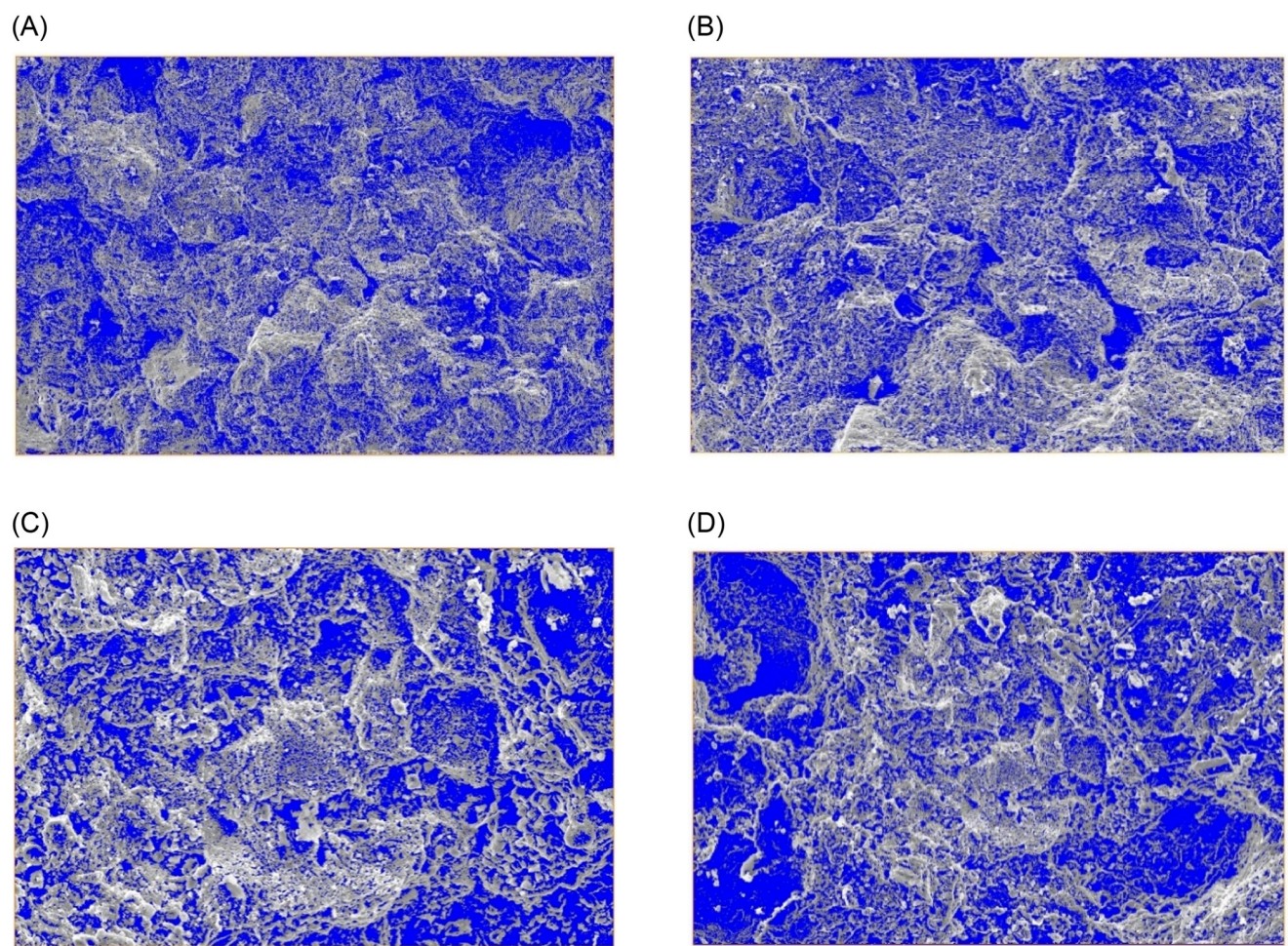

**Fig 4. Analysis of porosity in weakly cemented sandstone under different alkaline soaking conditions.** (A) Deionized distilled water. (B) 0.01mol/L $CaCl_2$ pH = 7. (C) 0.01mol/L $CaCl_2$ pH = 9. (D) 0.01mol/L $CaCl_2$ pH = 11.

**Table 4. Statistical distribution of microscopic pores in weakly cemented sandstone under different alkaline water conditions.**

| Parameter | | Deionized water | 0.01mol/L $CaCl_2$ pH = 7 Solution | 0.01mol/L $CaCl_2$ pH = 9 Solution | 0.01mol/L $CaCl_2$ pH = 11 Solution |
|---|---|---|---|---|---|
| **Porosity (%)** | | 29.09 | 31.91 | 33.62 | 40.39 |
| Pore size ($um$) | Mean | 2.05e-5 | 2.43e-5 | 2.64e-5 | 3.31e-5 |
| | Minimum | 6.03e-6 | 8.20e-6 | 1.03e-6 | 1.58e-6 |
| | Maximum | 0.0008 | 0.0011 | 0.0013 | 0.0023 |
| | Median | 1.36e-6 | 4.10e-6 | 5.15e-6 | 7.90e-6 |
| | Variance | 8.16e-10 | 1.25e-9 | 1.94e-9 | 6.89e-9 |
| Pore area ($um^2$) | | 2.16e-10 | 2.55e-10 | 3.06e-10 | 3.58e-10 |
| Roundness | Mean | 6.73 | 4.89 | 4.59 | 4.08 |
| | Minimum | 0.97 | 0.95 | 0.91 | 0.86 |
| | Maximum | 1094.45 | 925.00 | 763.00 | 613.00 |
| | Variance | 663 | 607 | 530 | 487 |

of 1.94e-9. The average pore area was 3.06e-10 $\mu m^2$. The roundness of the pores ranged from 0.91 to 763.00, with an average roundness of 4.59 and a variance of 530, indicating further dispersion in roundness data.

4. Under 0.01 mol/L $CaCl_2$ pH = 11 soaking conditions, the surface porosity reached 40.39%, representing a 38.84% increase relative to deionized distilled water and a 20.14% increase compared to 0.01 mol/L $CaCl_2$ pH = 9 soaking conditions. The pore diameters ranged from 1.58e-6 to 0.0023 $\mu m$, with an average of 3.31e-5 $\mu m$ and a variance of 6.89e-9. The average pore area was 3.58e-10 $\mu m^2$. The roundness of the pores ranged from 0.86 to 613.00, with an average roundness of 4.08.

To better understand the impact of porosity on pore characteristics, we analyzed the correlations between porosity and various parameters such as pore diameter, pore area, and roundness. The relationships are illustrated in Fig 5.

1. As porosity increases, the average pore diameter also increases. This indicates that as porosity grows, the size of the pores tends to become larger. Specifically, at higher porosity levels, the distribution of pore sizes is broader, and the mean value significantly rises. This trend suggests that the expansion of pores in an alkaline environment may promote the formation of larger pores.

2. There is a positive correlation between porosity and average pore area; the higher the porosity, the larger the average pore area. As porosity increases, the distribution of pore areas also tends to expand. This is because the alkaline environment promotes the expansion and connection of pores, increasing the area of individual pores. This finding is consistent with the increase in pore diameter, further indicating changes in pore structure under alkaline conditions.

3. As porosity increases, pore roundness tends to decrease. This suggests that as porosity grows, the shape of the pores becomes more elongated and less circular. Pores with higher porosity are closer to a circular shape, indicating that the alkaline environment may make the pore boundaries more uniform and smooth. This change in shape could be due to the intensified chemical reactions in the alkaline environment, leading to further erosion and adjustment of the pore boundaries.

Through the design of an equivalent factor formula, the number of elongated and round pores could be determined. Shape factors greater than 5 were considered elongated pores, while those less than 2 were considered round. In image processing, equivalent diameter is used to describe the size of irregular objects, so pore size distribution under different alkaline water conditions could be represented by a histogram of the equivalent radius of pores.

From Fig 6, it is evident that with increasing pH of the alkaline water, the number of elongated pores gradually decreases, while the number of round pores slowly increases. Under deionized water conditions, the equivalent diameter of pores ranged from 5.05e-6 to 3.69e-4 $u$m, with an average of 1.04e-5 $u$m. Under 0.01mol/L $CaCl_2$ pH = 7, the range was 2.02e-6 to 8.11e-6 $u$m, with an average of 4.24e-6 $u$m. Under pH = 9, it ranged from 5.05e-7 to 3.51e-5 $u$m, averaging 1.36e-6 $u$m, and under pH = 11, from 1.01e-6 to 7.39e-5 $u$m, averaging 2.69e-6 $u$m.

In summary, an enhancement in the alkalinity of solutions fosters a more intricate pore distribution within rocks, thereby progressively elevating the porosity of the rock mass [37, 38]. The mean pore diameter fluctuates in accordance with variations in the alkaline aqueous

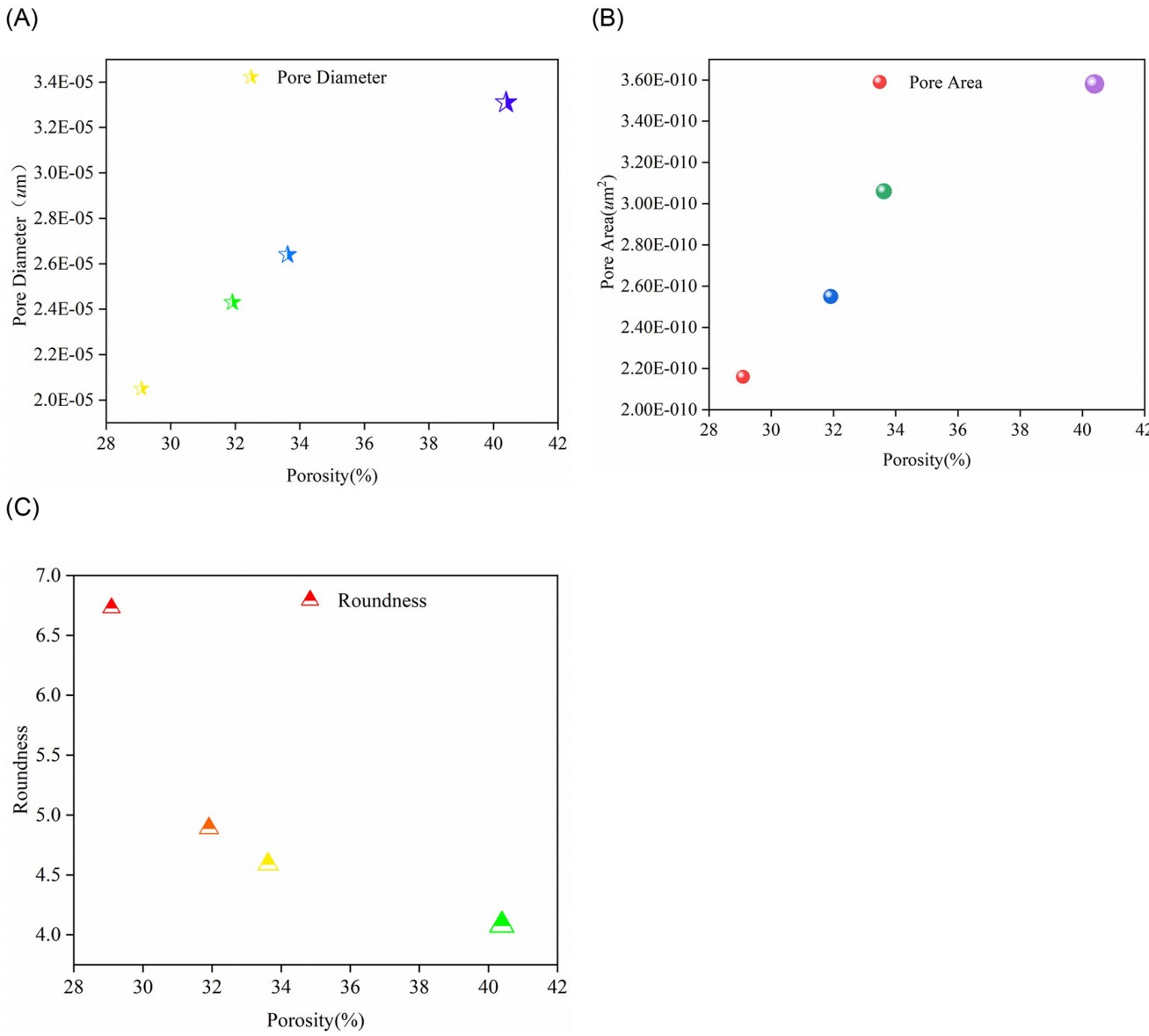

**Fig 5. The relationships between porosity and pore diameter, pore area, and roundness.** (A) The Relationship between Porosity and Pore Diameter. (B) The Relationship between Porosity and Pore Area. (C) The Relationship between Porosity and Roundness.

milieu, concurrently witnessing a reduction in the variance of pore sizes, indicative of an augmented uniformity in pore dimensions as the solution's alkalinity intensifies. This progression is accompanied by modifications in pore areas and circularity, underscoring a systematic transformation in pore morphology as alkalinity escalates.

A shift in pore shape factors occurs, manifesting a transformation in pore geometries; primitive elongated pores, subsequent to exposure to alkaline conditions, tend to transmute into elliptical and circular configurations. These observations highlight the pivotal role of alkaline fluids in mediating chemical reactions that bolster the stability of the rock structure, instigating structural alterations, mineral dissolution, and the genesis of novel clay minerals-collectively exemplifying a synergistic interplay between mineral dissolution and precipitation processes.

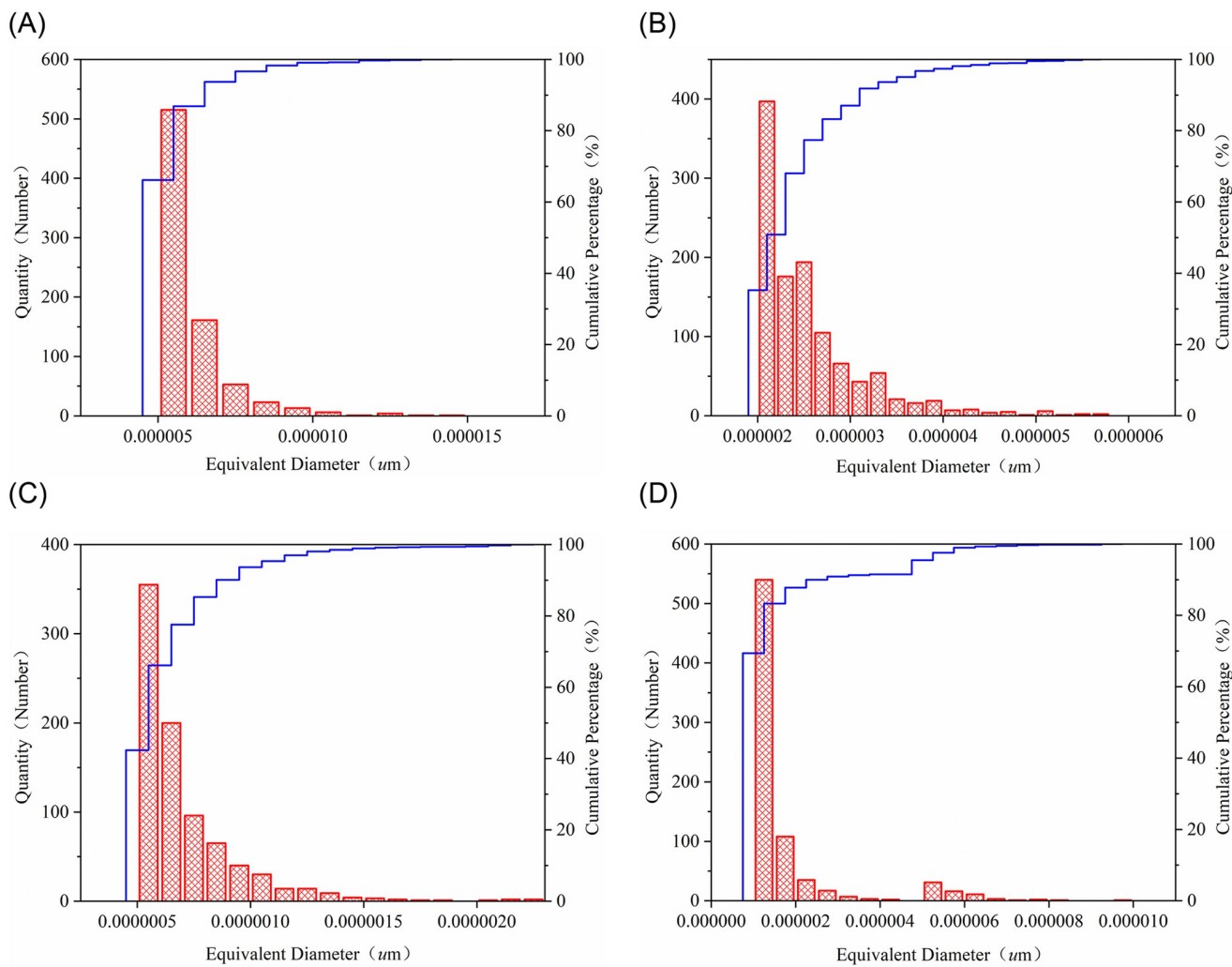

**Fig 6. Distribution of equivalent diameter of pores in weakly cemented sandstone under different alkaline soaking conditions.** (**A**) Deionized distilled water. (**B**) 0.01mol/L CaCl$_2$ pH = 7. (**C**) 0.01mol/L CaCl$_2$ pH = 9. (**D**) 0.01mol/L CaCl$_2$ pH = 11.

Furthermore, an upsurge in the pH of alkaline waters promotes the development of both primary and secondary porosities, augmenting porosity levels and prompting a morphological transition of pore shapes from elongated to elliptical and, ultimately, circular. Concomitantly, there is a diminution in the equivalent pore diameters, while circularity broadly exhibits a declining tendency, altogether emphasizing the profound and transformative influence of heightened alkalinity on the evolutionary trajectory of pore architectures within the lithic fabric.

## Changes in the pH of hydrochemical solutions

While analyzing the impact of alkaline solutions on rock mass damage, this study also explored the soaking effect of deionized distilled water, including four different pH hydrochemical solutions. During the experiment, the dynamic changes in the pH of the solutions were regularly measured and recorded using a pH pen. Fig 7 shows that the pH of the four solutions changed differently over time:

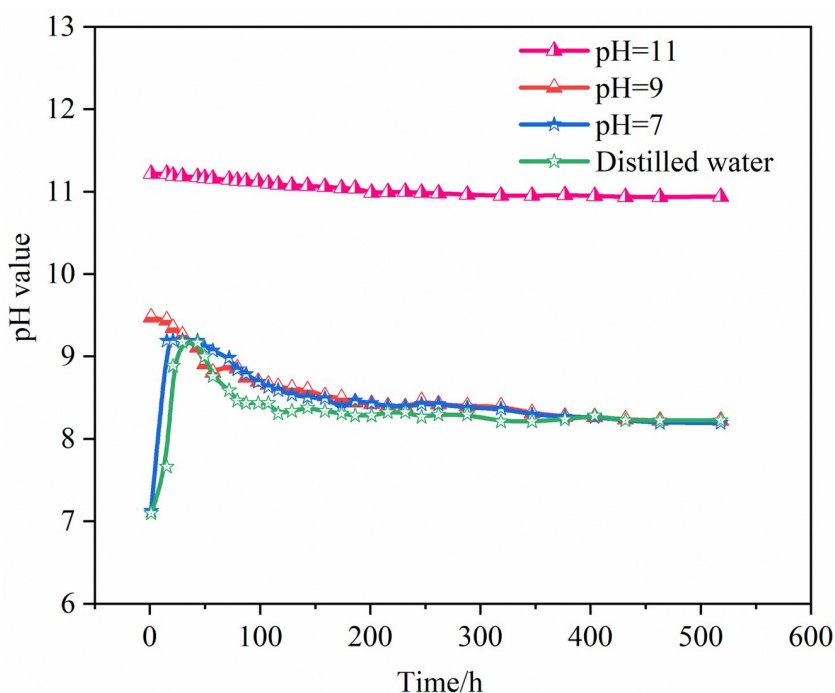

**Fig 7. Changes in pH over time in different hydrochemical solutions.**

1. For pH = 9 and 11 hydrochemical solutions, their pH decreased over time and eventually stabilized, reflecting the gradual consumption of alkaline components due to chemical reactions with the minerals in the rock, particularly the consumption of hydroxide ions in the solution. By contrast, the pH of deionized distilled water and pH = 7 initially increased, peaked, then gradually decreased, slowly stabilizing. This was due to initial chemical exchanges and reactions with the rock surface, but over time, the reactions balanced.

2. In the early soaking stage, the pH of deionized distilled water and pH = 7 solution fluctuated significantly within the first 48 hours, especially rising sharply in the first 24 hours to 9.22 and 9.17, respectively. The pH of pH = 9 solution initially decreased significantly in the first 48 hours, then gradually stabilized. These three solutions showed minor pH changes during the mid-soaking stage. For pH = 11 solution, although the pH continued to fluctuate, it remained within a stable range overall.

3. By 120 hours of soaking, the pH changes in the four solutions were minimal, maintaining a stable state. This was due to minerals such as quartz ($SiO_2$) and glauconite ($KAl_{3-}Si_3O_{10}(OH)_2$) in the solution undergoing hydrochemical reactions, producing $K^+$ and $Al^+$ ions that caused the original mineral structures to dissociate and release into the solution, binding with $OH^-$ ions, thus consuming a large amount of $OH^-$ ions and shifting the solution's pH towards weak alkalinity.

4. After 150 hours of soaking, due to the increased ion exchange and solubility between the weakly cemented sandstone and the hydrochemical solution, the decomposition of minerals like quartz consumed more $OH^-$ ions, causing the pH of the solution to decrease. Continuing to soak until 180 hours, as the ion exchange and hydrolysis reactions between the

rock and solution neared completion, the solution's pH stabilized and no longer exhibited significant changes.

## Uniaxial compression test results and analysis

**Failure modes of weakly cemented sandstone.**   Rock specimens soaked in 0.01mol/L $CaCl_2$ solution were used for rock mechanics experimental analysis. Prior to the experiments, the rock specimens that had been soaked were categorized and numbered for ease of subsequent testing and analysis, as shown in Table 5.

Subjected to external influences, the evolution of pores and fissures within rocks can significantly affect their mechanical properties and the underlying rules governing macroscopic damage, deformation, and failure behavior [39]. During uniaxial compression, the rock specimens exhibited three representative failure modes: conjugate oblique shear failure, single oblique shear failure, and tensile failure. Similarly, unsoaked specimens and those soaked in pH = 7, pH = 9, and pH = 11 solutions demonstrated these failure modes during uniaxial compression tests, as shown in Fig 8.

1. X-shaped Conjugate Shear Failure Mode: As illustrated in the figure, this failure mode is particularly prominent in an alkaline aqueous immersion environment. The alkaline medium accelerates the dissolution of bonding materials within the rock, thereby exacerbating variations in its microstructure. In this process, two intersecting fracture planes form, creating an oblique angle with respect to the direction of applied load, directly illustrating the outcome where shear stress surpasses the rock's bearing threshold. With sustained external loading, microcracks within the rock gradually extend and interconnect, culminating in the characteristic X-shaped fracture pattern. This evolution underscores the complex interplay between shear forces and changes in the rock's internal structure.
The rock samples display a propensity for plastic flow behavior, evidenced by a rise in internal plastic deformation that, in turn, diminishes the rock's comprehensive strength. Initiation of the failure progression is gradual and continuous, departing from typical brittle fracture modes and exhibiting heightened plastic deformation traits. Under these circumstances, crack propagation assumes a more complex and refined pattern, indicative of sophisticated fracturing dynamics.

**Table 5. Soaking conditions of rock specimens under different conditions.**

| Serial Number | pH Value | Sample Number | Number of Specimen | Remarks |
|:---:|:---:|:---:|:---:|:---:|
| 1 | No corrosion | n-1 | 3 | Standard Specimen |
| 2 | | n-2 | | |
| 3 | | n-3 | | |
| 4 | 7 | 7–1 | 3 | Deionized Water |
| 5 | | 7–2 | | |
| 6 | | 7–3 | | |
| 7 | 9 | 9–1 | 3 | |
| 8 | | 9–2 | | |
| 9 | | 9–3 | | |
| 10 | 11 | 11–1 | 3 | |
| 11 | | 11–2 | | |
| 12 | | 11–3 | | |

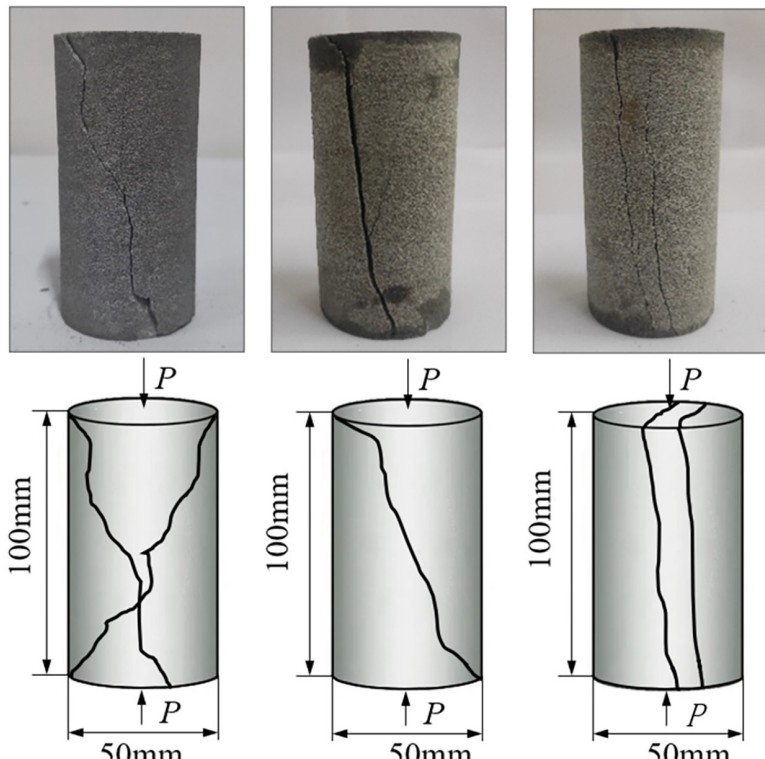

**Fig 8. Shapes and schematic diagrams of uniaxial compression test specimens under stress.** (A) Conjugate oblique shear failure. (B) Single oblique shear failure. (C) Tensile failure.

2. Single-plane Shear Failure Mode: Under identical alkaline water soaking conditions, this failure mode exhibits its distinctiveness, characterized by the presence of a single dominant shear fracture plane. This type of failure arises from the accumulation of shear stress on a specific plane exceeding the rock's shear strength limit, embodying the classic shear failure mechanism. Of note, the maximum resistance before shear failure is not only dependent on the magnitude of shear stress but is also intricately linked to the normal stress acting on that plane, reflecting an adjustment of the rock's shear strength under the combined influence of normal and shear stresses. Thus, it can also be regarded as a form of combined compressive-shear failure. Unlike the X-shaped conjugate shear, this mode highlights scenarios where stress in a single direction overwhelms the rock's shear resistance.

3. Tensile Failure Mode: In the single-axis compression testing under alkaline water conditions, the tensile failure of rocks is equally remarkable. Transverse tensile stress induced by axial pressure, when surpassing the rock's tensile strength, leads to prominent radial fractures. The alteration of the rock's internal structure due to alkaline immersion enhances its susceptibility to lateral stress, facilitating the occurrence of tensile failure. This phenomenon emphasizes how environmental factors, by altering the internal stress state of the rock, facilitate the development of specific failure mechanisms.

Through these systematic descriptions, our understanding of rock failure modes under specific environmental loads is not only enhanced but also provides a firmer theoretical foundation for subsequent mechanical analyses and predictions.

**Analysis of the failure process of weakly cemented sandstone.** Unsoaked specimens and those soaked in pH = 7, pH = 9, and pH = 11 solutions were subjected to uniaxial compression tests to analyze the mechanical properties of the rock at various stages, as shown in Fig 9. In each hydrochemical environment, the rock's stress-strain curve can be divided into four stages: compaction, elasticity, yield, and post-yield.

Compaction Stage: Under axial pressure, the rock's original open structural planes and microcracks gradually close, causing compaction. In this stage, strain increases rapidly, while stress growth is relatively slower. As the soaking environment's pH increases, the duration of the rock's uniaxial compression compaction stage tends to extend. This is due to the dissolution of minerals and changes in cementing material properties caused by alkaline water,

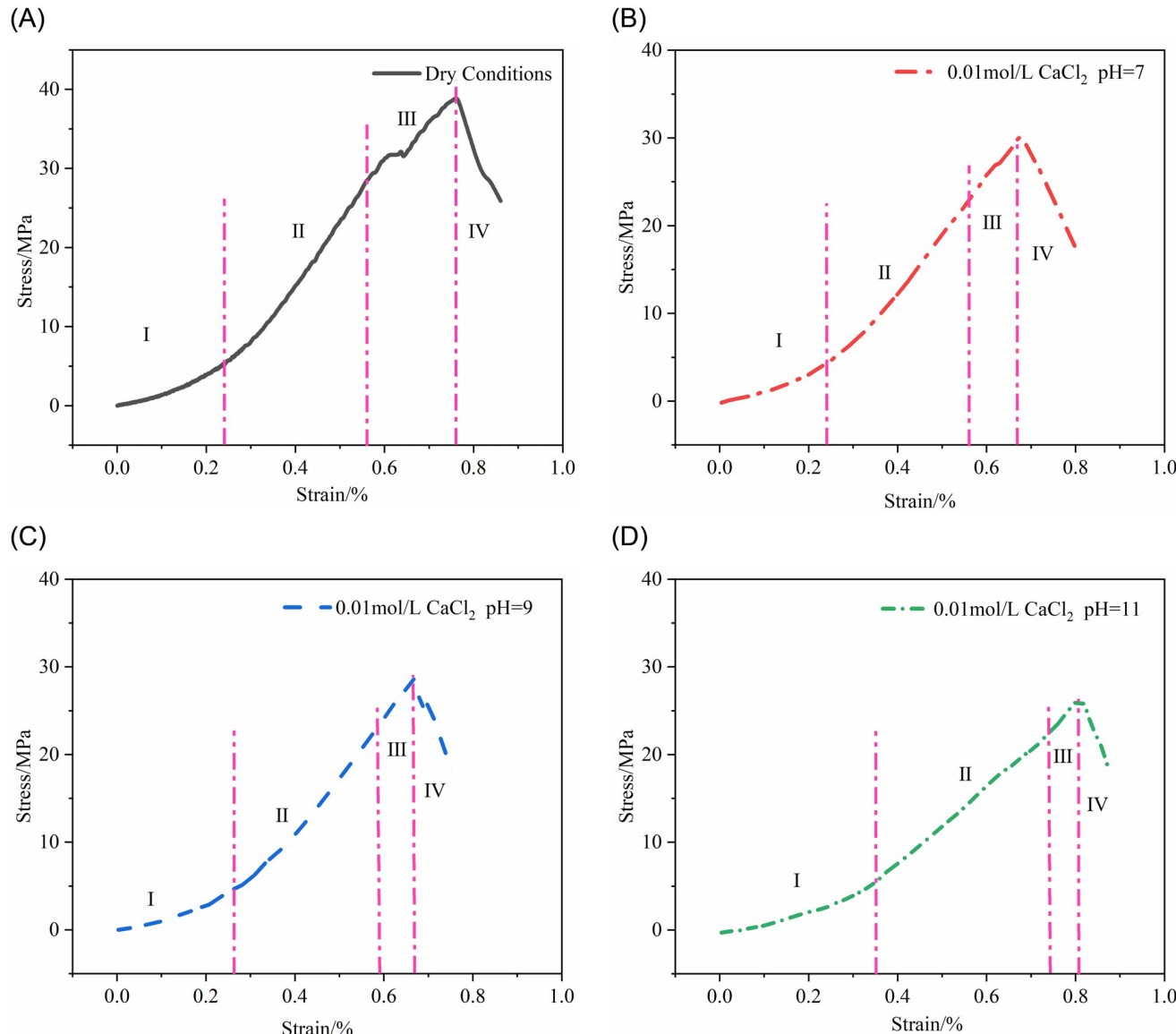

**Fig 9. Stress-strain curves of rock specimens under uniaxial compression in different soaking conditions.** (A) Dry state. (B) pH = 7 CaCl₂ soaking solution. (C) pH = 9 CaCl₂ soaking solution. (D) pH = 11 CaCl₂ soaking solution.

expanding the rock's pore structure and forming more cracks and holes, affecting the rate of internal structure closure.

Elastic Stage: The curve is almost linear, with stress and strain growing linearly and uniformly. Microcracks continue to close, and due to the reduction of cementing substances, new microcracks easily form, leading to fracture. The increasing pH of alkaline water significantly affects the amount of stress and strain change in this stage, reducing the amount of stress change. The slope, representing the modulus of elasticity, shows Dry (4.7GPa) > pH = 7 (4.4GPa) > pH = 9 (3.8GPa) > pH = 11 (3.1GPa).

Yield Stage: The rock specimens enter an irreversible stage, exhibiting plastic deformation and crack expansion. The stress-strain curve shows non-linear characteristics. With increasing pH of alkaline water, the rock's strain values correspondingly increase. Peak strength is Dry (38.8MPa) > pH = 7 (30.1MPa) > pH = 9 (28.7MPa) > pH = 11 (26.7MPa), while peak strain values follow the order of Dry < pH = 7 < pH = 9 < pH = 11.

Post-Yield Stage: After the rock specimens reach peak strength, their internal structure is damaged, but they largely maintain their overall shape. Due to stress concentration and deformation, existing cracks in the rock further expand, forming along existing microcracks or in new areas, further damaging the rock's internal structure. Under alkaline water soaking, changes in the properties of cementing substances weaken the bond between rock particles, leading to displacement and relative motion of micro-particles, and eventually particle fragmentation. As cracks expand and particle fragmentation increases, the rock's overall structure gradually becomes unstable, leading to a decline in strength, but not to zero, indicating that fractured rock still has some load-bearing capacity.

**Microstructural changes in the elastic modulus of weakly cemented sandstone in alkaline environments.** The modulus of elasticity is a key indicator of a rock's resistance to deformation, directly reflecting its internal structure's rigidity [40, 41]. In alkaline environments, changes in the microstructural porosity and fracture structure due to chemical solutions' action are crucially important to the modulus of elasticity. With increasing pH of the solution, the microstructure of weakly cemented sandstone specimens undergoes disintegration, with cementing materials like silicates and carbonates dissolving faster in alkaline water solutions with pH greater than 9, leading to a decrease in the degree of cementation. This process not only increases the number and volume of pores but may also induce the formation of new microcracks or the expansion of existing ones, thereby weakening the rock's cohesion and structural integrity.

Under neutral conditions (pH = 7), the cementing materials of weakly cemented sandstone are relatively stable, with minimal impact on pores and fractures, thus maintaining a higher modulus of elasticity. However, with increasing alkalinity of the solution, especially when the pH exceeds 9, the dissolution of cementing materials leads to significant changes in the pore structure, increasing the number and connectivity of microcracks, reducing the rock's load transfer ability, thereby decreasing its resistance to compressive forces, as reflected in the reduced modulus of elasticity. In summary, as the pH of the solution increases, the modulus of elasticity of weakly cemented sandstone specimens gradually decreases, as shown in Fig 10.

## Conclusion

This study has thoroughly investigated the impact of alkaline solutions on the mass damage and mechanical properties of weakly cemented sandstone. The primary findings of this research are as follows:(1) The experiments indicate that as the alkalinity (pH value) of the soaking solution increases, the degree of mass damage to the rock significantly escalates.

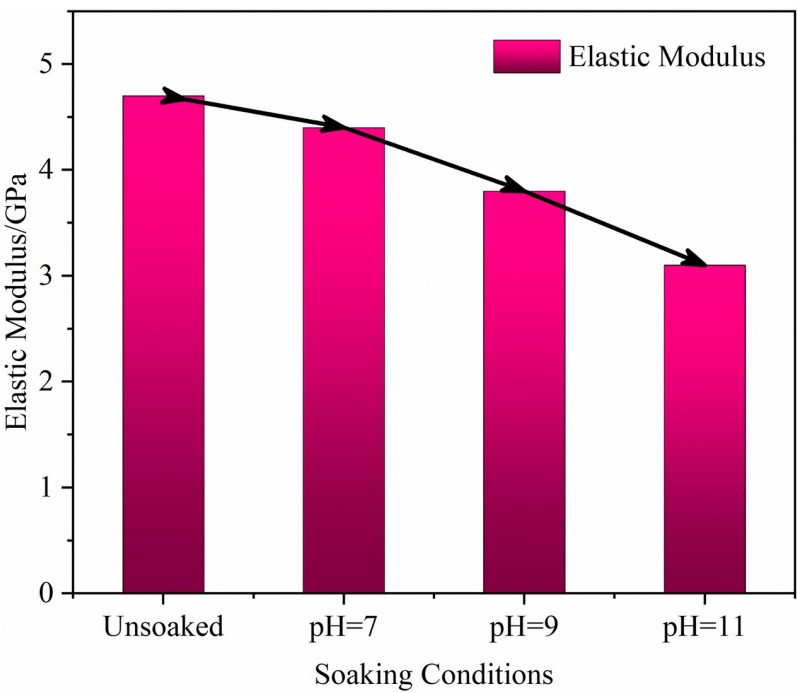

**Fig 10. Analysis of rock modulus of elasticity under different soaking conditions.**

Particularly when the pH value reaches or exceeds 9, complexes composed of cations like $Ca^{2+}$ and $Mg^{2+}$ can block the pores and fractures in the rock. The protective layers formed by these complexes effectively prevent further infiltration of moisture and chemicals, thereby significantly slowing down additional mass damage to the rock. This discovery highlights the crucial role of cations in maintaining rock structural stability in alkaline environments.(2) The microscopic pore analysis indicates that as the alkalinity of the solution increases, there is a continuous increase in porosity and a corresponding decrease in the variance of pore sizes, leading to a greater uniformity among the pore sizes. The original elongated pores at the edges lose their cementation, altering their shape towards more elliptical and circular forms. The primary pores continue to expand slowly, while new pores further develop. The evolution of pores is increasingly away from elongated forms, progressing towards elliptical and even circular shapes.(3) The study found that the pH of the alkaline solution exhibits a decreasing trend over time due to the consumption of alkaline components in the solution. In contrast, the pH of deionized distilled water and solutions with a pH of 7 initially rises, then gradually decreases, and eventually stabilizes. This change indicates that the reaction between the alkaline solution and minerals leads to the consumption of hydroxide ions, and this reaction stabilizes over time.(4) After hydrochemical corrosion, the uniaxial compressive strength of weakly cemented sandstone specimens decreases with increasing pH of the alkaline solution, while the modulus of elasticity shows an increasing trend with increasing pH. This trend reflects the complex response of rock structures to alkaline solutions and reveals the intricate impact of alkaline solutions on the corrosion of rock minerals and internal fracture changes. These studies not only enhance our understanding of the mechanical behavior of rocks in alkaline environments but also provide important guidance for engineering practice management in related fields.

## Supporting information

**S1 File.**
(DOCX)

## Acknowledgments

The authors would like to thank the staff of the Radiology Department at Daxing Hospital.

## Author Contributions

**Conceptualization:** Xiaoshi Li.

**Data curation:** Qingsong Zhuo, Qian Zheng, Mingang Zhang, Xiaoyu Zhao, Jigang Geng, Xiaoshi Li, Ruoyu Bao.

**Formal analysis:** Jigang Geng, Ruoyu Bao.

**Investigation:** Qian Zheng, Mingang Zhang, Xiaoyu Zhao, Xiaoshi Li, Ruoyu Bao.

**Methodology:** Qingsong Zhuo, Qian Zheng, Bin Wang, Mingang Zhang, Xiaoyu Zhao, Jigang Geng.

**Resources:** Jie Zhang, Qingsong Zhuo, Qian Zheng, Ruoyu Bao.

**Software:** Qingsong Zhuo, Mingang Zhang, Xiaoyu Zhao, Jigang Geng, Xiaoshi Li.

**Writing – original draft:** Jie Zhang, Qingsong Zhuo.

**Writing – review & editing:** Jie Zhang, Qingsong Zhuo.

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
