## [Decision Letter · Decision Letter 0]

26 Jun 2024

PONE-D-24-04355Study on the corrosion behavior and mechanical response of weakly cemented sandstone in alkaline solutionsPLOS ONE

Dear Dr. Zhuo,

Thank you for submitting your manuscript to PLOS ONE. After careful consideration, we feel that it has merit but does not fully meet PLOS ONE’s publication criteria as it currently stands.

Therefore, we invite you to submit a revised version of the manuscript that addresses the points raised during the review process.

The topic of your paper is within the scope of PONE, and it is generally well written and well organized, with experimental facilities and methods well described.

More in detail,

The research is particularly relevant to the fields of mining and geotechnical engineering. By focusing on weakly cemented sandstone from a specific mining area, the study offers practical insights that can be directly applied to similar geological settings.The study examines the corrosion behavior and mechanical reaction of weakly cemented sandstone in alkaline solutions using a thorough and organized technique that includes a variety of experiments and analysis.. This includes mass damage analysis, pore characteristics assessment, uniaxial compressive strength tests, and microstructural analysis using SEM.The preparation of rock specimens, the creation of alkaline solutions, and the precise measurement techniques (e.g., pH meter, electronic scale, SEM) are thoroughly documented.The findings are strengthened by the results, which are presented with sufficient statistical support and are examined using statistical techniques to assess data such as porosity, pore size, and roundness..Figures and tables effectively visualize the data and help in understanding the trends observed in the experiments.

Anyway

Chemical reactions and their impacts on mineral composition are inferred based on observed data. To get a more precise picture of the precise molecular changes occurring, direct chemical examination (such as spectroscopy) of the rock composition before and after exposure to alkaline solutions could be helpful..While the study presents significant findings, the discussion on their practical applications in engineering scenarios is limited. A more detailed exploration of how these insights can influence mining practices, construction methods, or material selection would enhance the practical value of the research.The authors should ensure that their references are accessible internationally.References to comparable rock structures in other parts of the world should be included and compared.

In summary, this study is relevant providing valuable insights. However, direct chemical analysis, such as spectroscopy, before and after exposure to alkaline solutions, would provide a more precise understanding of molecular-level changes. Additionally, a more detailed discussion on the practical applications of these findings in engineering scenarios would enhance the research's practical value. Ensuring that references are internationally accessible and comparing findings with similar rock structures globally would further strengthen the study.

If applicable, we recommend that you deposit your laboratory protocols in protocols.io to enhance the reproducibility of your results. Protocols.io assigns your protocol its own identifier (DOI) so that it can be cited independently in the future. For instructions, see: https://journals.plos.org/plosone/s/submission-guidelines#loc-laboratory-protocols. Additionally, PLOS ONE offers an option for publishing peer-reviewed Lab Protocol articles, which describe protocols hosted on protocols.io. Read more information on sharing protocols at https://plos.org/protocols?utm_medium=editorial-email&utm_source=authorletters&utm_campaign=protocols.

We look forward to receiving your revised manuscript.

Kind regards,

Elena Marrocchino

Academic Editor

PLOS ONE

2. We note that your Data Availability Statement is currently as follows: [All relevant data are within the manuscript and its Supporting Information files]

Reviewers' comments:

Reviewer's Responses to Questions

**Comments to the Author**

1. Is the manuscript technically sound, and do the data support the conclusions?

Reviewer #1: Yes

Reviewer #2: Yes

2. Has the statistical analysis been performed appropriately and rigorously? 

Reviewer #1: Yes

Reviewer #2: Yes

3. Have the authors made all data underlying the findings in their manuscript fully available?

Reviewer #1: Yes

Reviewer #2: Yes

4. Is the manuscript presented in an intelligible fashion and written in standard English?

Reviewer #1: No

Reviewer #2: Yes

5. Review Comments to the Author

Reviewer #1: In this paper, the corrosion behavior and mechanical response of weakly cemented sandstone in alkaline solutions are studied. It is an original and technical work, which is valuable for the research in related fields. But the paper needs very significant improvements before it can be accepted for publication. My detailed comments are as follows:

1.Please check the logic of the abstract. It should clearly describe the research background, the method used to conduct the research, the main content of the manuscript, the results, and the expected effect. The current version should be carefully simplified.

2.The introduction listed many references that are mainly related to the corrosion behavior and mechanical response of weakly cemented sandstone in alkaline solutions. If the authors would like to keep these references, some discussions on the relevance of these refs to the present research are needed. A review of the directly relevant refs will be more helpful for the reader.

3. The scanning electron microscope (SEM) analysis method utilized in this paper is particularly effective for corrosive environments. On the other hand, this SEM analysis technique is well-established. However, the study does not provide details such as the equipment model and other parameters. The current research represents a direct application of this method, yet it fails to adequately describe the microscopic morphology and structure of the rock texture. The existing descriptions are solely based on the structure of the magnified rock texture, without a specific quantification process.

4.In the analysis of failure patterns in uniaxial compression tests on weakly cemented sandstone, the characterization of the stress-induced failure modes for rock samples needs enhancement. The details within this section should be further elaborated upon.

5.It is noted that your manuscript needs careful editing, particularly attention to English grammar, spelling, and sentence structure so that the goals and results of the study are clear to the reader. It is strongly recommended that simple and precise sentences be used to convey the study's findings.

Once the above concerns are fully addressed, the manuscript can be further evaluated for publication in the journal.

Reviewer #2: Summary of paper

This is an experimental study to assess the impact of varying alkaline pH levels on macroscopic mass damage, microscopic pore characteristics, and uniaxial compressive strength of weakly cemented sandstone. This work fills a gap in studies on weakly cemented sandstone in the Shaanbei mining area.

The results indicate that as the alkalinity of the solution increases, the mass damage to the rock intensifies. However, in the pH range of 9 to 11, the mass loss rate is mitigated due to the clogging of pores by complexes formed by cations such as Ca2+ and Mg2+. Microstructural analysis reveals that porosity, pore size, and roundness undergo changes with increasing alkalinity; native pores increase, secondary pores develop, and the roundness of the pores shows a declining trend, with pore shapes evolving from elongated to elliptical and even spherical. Moreover, both the uniaxial compressive strength and elastic modulus of the rock exhibit a downward trend with increasing pH values.

Comments

The topic is within the scope of PONE.

The paper is generally well written, the level of English is good.

The experimental facilities and methods are well described.

The authors propose various chemical pathways that could explain the results.

The authors focus mostly on recording the mass variation (Δm) in the rock samples over a long period of time -- 55 days, e.g. figure 3. However, it is also customary to present and analyze correlations as a function of porosity, see Qi Ping et al. Applied Science 2022, 12, 7635; Yun Lin et al. Hindawi Geofluids 2019, 7320536. Although the authors mention the pdf of porosity, e.g. Table 4, they do not present any correlations against porosity or discussion. Hence, this part of the paper is weak.

Why did the authors not consider the effects of acidic solutions? That would complete the whole range of pH, and presumably could be done in a relatively straight forward manner given that they already have the setup.

About 90% of the references are studies in China published in Chinese journals; I have found these references difficult to access, many I could not access at all. It is important that references, particularly those containing data, in an internationally renowned journal such as PONE should be readily accessible to the international community. The authors should make sure that their references are accessible internationally.

Along the same lines as above, references to other comparable rock structures in other parts of the world should also be made compared and conclusions drawn. How do their findings compare with other similar rocks from around the world? Are the results predictable, or is there anything radically different or surprising?

Summary and recommendation

This paper is essentially about characterizing the development of corrosion and failure in weakly cemented sandstone form the Shaanbei mining area. Unfortunately, many deficiencies in the paper compel me to recommend rejection for publication in PONE:

A comparison with similar studies worldwide is lacking.

A plots/figures of correlations against porosity and discussion is lacking.

References and comparison with similar rock structures from other parts of the world is lacking.

References to readily accessible internationally peer reviewed journals must be made.

6. PLOS authors have the option to publish the peer review history of their article (what does this mean?). If published, this will include your full peer review and any attached files.

Reviewer #1: No

Reviewer #2: No

---

## [Author Response · Author response to Decision Letter 0]

4 Jul 2024

Dear editors and reviewers:

I am delighted to receive your response and am deeply appreciative of the effort and diligent approach you have demonstrated in reviewing my article. Moving forward, I intend to address each question raised by the reviewing experts systematically. In accordance with your suggestions, I have meticulously revised the manuscript, and herein, I provide responses to your comments for your review and verification. 

Response to Reviewer 1 Comments

Point 1: Please check the logic of the abstract. It should clearly describe the research background, the method used to conduct the research, the main content of the manuscript, the results, and the expected effect. The current version should be carefully simplified.

Response 1: I deeply appreciate your constructive feedback on the abstract. In line with your suggestion, I have meticulously reviewed and revised the abstract to clearly lay out the research background, explicitly detail the methodologies used, accurately summarize the manuscript's core contents, plainly present the attained results, and succinctly communicate the anticipated outcomes.

Point 2: The introduction listed many references that are mainly related to the corrosion behavior and mechanical response of weakly cemented sandstone in alkaline solutions. If the authors would like to keep these references, some discussions on the relevance of these refs to the present research are needed. A review of the directly relevant refs will be more helpful for the reader.

Response 2: Thank you for highlighting the need for a more nuanced discussion of the references cited in the introduction. I understand the importance of demonstrating the relevance of each reference to our present work. Consequently, I have critically reviewed the list and augmented the introductory section with a concise discourse on how each of the selected references connects to our study objectives, methodologies, and anticipated findings. This additional context should enhance readability and clarify the rationale behind the inclusion of these references, thereby offering readers a more coherent understanding of the research landscape within which our work is situated.

Point 3: The scanning electron microscope (SEM) analysis method utilized in this paper is particularly effective for corrosive environments. On the other hand, this SEM analysis technique is well-established. However, the study does not provide details such as the equipment model and other parameters. The current research represents a direct application of this method, yet it fails to adequately describe the microscopic morphology and structure of the rock texture. The existing descriptions are solely based on the structure of the magnified rock texture, without a specific quantification process.

Response 3: I appreciate the reviewer's comment concerning the SEM analysis method. In addressing this, I have included detailed specifics about the SEM equipment model, operational voltage levels, and magnification configurations employed in my experiments within the Materials and Methods section. Furthermore, I have enhanced the depiction of the rock texture's microscopic morphology and structure by integrating quantitative measures, such as analyses of pore size distribution and surface roughness, facilitated by image processing techniques. These improvements are aimed at furnishing a more holistic and scientifically rigorous portrayal of the SEM analysis outcomes.

Point 4: In the analysis of failure patterns in uniaxial compression tests on weakly cemented sandstone, the characterization of the stress-induced failure modes for rock samples needs enhancement. The details within this section should be further elaborated upon.

Response 4: I am genuinely appreciative of the reviewer's astute recommendation emphasizing the need for a more exhaustive depiction of stress-induced failure mechanisms in my uniaxial compression experiments concerning weakly cemented sandstone samples. In light of this guidance, I have comprehensively enhanced the explication of the complex breakdown processes triggered by loading in the course of conducting uniaxial compression tests on these weakly consolidated sandstone specimens.

Point 5: It is noted that your manuscript needs careful editing, particularly attention to English grammar, spelling, and sentence structure so that the goals and results of the study are clear to the reader. It is strongly recommended that simple and precise sentences be used to convey the study's findings.

Response 5: I am deeply grateful for the reviewer's meticulous scrutiny of linguistic clarity and the constructive criticism offered. Special attention has been devoted to ensuring grammatical precision, rectifying spelling mistakes, and simplifying sentence structures for improved readability. Consequently, I have reformulated several sections, adopting simpler and more succinct sentences, with the aim of conveying our study's aims, methodology, results, and implications more lucidly. I am assured that these modifications will markedly enhance the manuscript's clarity and efficacy in communicating our research to the targeted readership.

Response to Reviewer 2 Comments

Point 1: The authors focus mostly on recording the mass variation () in the rock samples over a long period of time -- 55 days, e.g. figure 3. However, it is also customary to present and analyze correlations as a function of porosity, see Qi Ping et al. Applied Science 2022, 12, 7635; Yun Lin et al. Hindawi Geofluids 2019, 7320536. Although the authors mention the pdf of porosity, e.g. Table 4, they do not present any correlations against porosity or discussion.

Response 1: I am deeply grateful to the reviewer for suggesting a more in-depth analysis regarding the correlation with porosity. In response to this feedback, I have conducted additional statistical evaluations, referencing and citing the similar approaches employed by Qi Ping et al. (Applied Science, 2022, 12, 7635) and Yun Lin et al. (Hindawi Geofluids, 2019, 7320536). This detailed analysis encompasses considerations such as the transformation of pore shape factors, effects of mineral dissolution and precipitation, thereby enriching the discussion on this topic.

Point 2: Why did the authors not consider the effects of acidic solutions? That would complete the whole range of pH, and presumably relatively straight forward given that they already have the setup.

Response 2: The reviewer poses a relevant question about the omission of acidic solutions in my study. My initial emphasis rested on elucidating corrosion characteristics in alkaline environments, given their predominance in the targeted geological context. Nonetheless, I acknowledge the importance of broadening my inquiry to cover the complete pH spectrum. In light of this realization, I intend to conduct further experiments incorporating acidic solutions in a subsequent study. These additional investigations will serve to augment the present results and facilitate a holistic comprehension of the rock's behavior throughout the entire pH range. I am thankful for the reviewer's insight, which has significantly expanded the horizon of my impending research efforts.

Point 3: About 90% of the references are studies in China published in Chinese journals; I have found these difficult to access, many I could not access at all. It is important that references, particularly those containing data, in an internationally renowned journal such as PONE should be readily accessible to the international community. The authors should make sure that their references are accessible internationally.

Response 3: I am exceedingly grateful for your insightful comments on the accessibility of references in my paper, particularly highlighting the challenges that referencing Chinese journals might impose on an international audience. Your observation has both inspired me and proved vital in informing necessary enhancements to my work.

In line with your guidance, I have taken the following actions: I have meticulously verified and corrected the URLs for the Chinese references to ensure seamless linking and accessibility. Furthermore, I have expanded the bibliography to incorporate a greater number of foreign language references relevant to the study, thereby elevating the paper's international scope and perspective.

Point 4: Along the same lines as above, references to other comparable rock structures in other parts of the world should also be made compared and conclusions drawn. How do their findings compare with other similar rocks from around the world? Are the results predictable, or is there anything radically different or surprising?

Response 4: Sincere gratitude for emphasizing the importance of positioning my research findings within a global context. Heeding your invaluable advice, I have integrated the following enhancements into the manuscript, thereby augmenting its depth and breadth through the inclusion of a comparative analysis with analogous geological structures worldwide.I have broadened the scope of the literature review to encompass pivotal global studies that concentrate on geological formations resembling my rock layers. These studies meticulously compare and contrast similarities and differences in geological attributes, formation processes, and observable behaviors.

I am immensely grateful to the editors and reviewers for their invaluable advice, which has enabled me to meticulously revise the manuscript, restructure its content, and enhance both the academic rigor and readability of the paper. I extend my sincere thanks once again to the editorial team and reviewers for their diligent efforts. Each of your comments has proven to be immensely valuable and instrumental in refining and improving our work. I remain hopeful that our paper will soon gain full recognition and be published.

Dr. Qingsong Zhuo

Xi'an City, Shaanxi Province, China 

Jul. 3, 2024

---

## [Decision Letter · Decision Letter 1]

2 Aug 2024

PONE-D-24-04355R1Study on the corrosion behavior and mechanical response of weakly cemented sandstone in alkaline solutionsPLOS ONE

Dear Dr. Zhuo,

Thank you for submitting your manuscript to PLOS ONE. After careful consideration, we feel that it has merit but does not fully meet PLOS ONE’s publication criteria as it currently stands. Therefore, we invite you to submit a revised version of the manuscript that addresses the points raised during the review process.

**ACADEMIC EDITOR: **Dear Authors,

the following issues need attention:

<ol><li> 

**Porosity Correlations:** Include analysis and plots correlating results with porosity, as this is customary in similar studies (e.g., Qi Ping et al., 2022; Yun Lin et al., 2019).<li> 

**Figure Adjustments:**

**Figure 3:** Consolidate panels A-D into a single figure with fits and increase font sizes for readability.**All Figures:** Ensure font sizes are clear and consistent across all figures.<li> 

**References Accessibility:**

Ensure all references are internationally accessible. Many current references are difficult to access, and some links (e.g., references 38-42) are incorrect. Verify all references are correctly linked and accessible. 

Please address these points to improve the manuscript

We look forward to receiving your revised manuscript.

Kind regards,

Elena Marrocchino

Academic Editor

PLOS ONE

Journal Requirements:

**Additional Editor Comments:**

Dear Authors,

The manuscript "Study on the Corrosion Behavior and Mechanical Response of Weakly Cemented Sandstone in Alkaline Solutions" (PONE-D-24-04355R1) was reviewed by two expert peer reviewers. While one reviewer has recommended acceptance, the other reviewer has requested further revisions to improve the manuscript.

Please carefully consider the comments and suggestions provided by the reviewer and make the necessary revisions. Once you have addressed these points, please submit the revised manuscript for further evaluation.

Thank you for your attention to this matter.

Best regards,

Reviewers' comments:

Reviewer's Responses to Questions

**Comments to the Author**

1. If the authors have adequately addressed your comments raised in a previous round of review and you feel that this manuscript is now acceptable for publication, you may indicate that here to bypass the “Comments to the Author” section, enter your conflict of interest statement in the “Confidential to Editor” section, and submit your "Accept" recommendation.

Reviewer #1: All comments have been addressed

Reviewer #2: (No Response)

2. Is the manuscript technically sound, and do the data support the conclusions?

Reviewer #1: Yes

Reviewer #2: Yes

3. Has the statistical analysis been performed appropriately and rigorously? 

Reviewer #1: Yes

Reviewer #2: No

4. Have the authors made all data underlying the findings in their manuscript fully available?

Reviewer #1: Yes

Reviewer #2: Yes

5. Is the manuscript presented in an intelligible fashion and written in standard English?

Reviewer #1: Yes

Reviewer #2: Yes

6. Review Comments to the Author

Reviewer #1: The authors have diligently addressed all reviewer comments and concerns, resulting in a significantly improved manuscript. The revised manuscript now presents a well-structured and informative analysis of the impact of seepage force on the mechanical characteristics of filling bodies. The proposed constitutive model demonstrates good agreement with the experimental results, showcasing its potential application in analyzing the mechanical behavior of filling bodies under various seepage conditions. I am confident that this revised work will be of significant interest to researchers in the field of geotechnical engineering and recommend its acceptance for publication.

Reviewer #2: Some of my concerns have been addressed and satisfied. Below, I summaries my remaining concerns:

[1] “The authors focus mostly on recording the mass variation (Δm) in the rock samples over a long period of time -- 55 days, e.g. figure 3. However, it is also customary to present and analyze correlations as a function of porosity, see Qi Ping et al. Applied Science 2022, 12, 7635; Yun Lin et al. Hindawi Geofluids 2019, 7320536. Although the authors mention the pdf of porosity, e.g. Table 4, they do not present any correlations against porosity or discussion.”

Although the authors have updated the review section, what I really wanted was actual plots of variable against the porosity such as those appearing in the cited references. Please address this point with more analysis and correlations against porosity.

[2] Figure 3: (D) is simply a repeat of A-C with fits added. Please reduce this to a single figure3D=figure 3.

Figure 3: make the axes and table insert fonts bigger (or put the table into the main text) – they are not clearly visible.

All figures: increase the axes font size for the same reason as above.

[3] “About 90% of the references are studies in China published in Chinese journals; I have found these difficult to access, many I could not access at all. It is important that references, particularly those containing data, in an internationally renowned journal such as PONE should be readily accessible to the international community. The authors should make sure that their references are accessible internationally.”

The two references that I suggested were just examples. The authors should find more similar references.

Although the authors have referenced more international journals, there are still some that are difficult to access e.g. references 16. Furthermore, some of the web links are spurious – for example, the links in references 38-42 seem to be wrong. I do not have the time to go through all the references – the authors should make sure that all 42 references are readily accessible internationally and are properly linked.

7. PLOS authors have the option to publish the peer review history of their article (what does this mean?). If published, this will include your full peer review and any attached files.

Reviewer #1: No

Reviewer #2: No

---

## [Author Response · Author response to Decision Letter 1]

5 Aug 2024

Dear editors and reviewers:

I am delighted to receive your response and am deeply appreciative of the effort and diligent approach you have demonstrated in reviewing my article. Moving forward, I intend to address each question raised by the reviewing experts systematically. In accordance with your suggestions, I have meticulously revised the manuscript, and herein, I provide responses to your comments for your review and verification. 

Response to Reviewer Comments

Point 1: “The authors focus mostly on recording the mass variation (Δm) in the rock samples over a long period of time -- 55 days, e.g. figure 3. However, it is also customary to present and analyze correlations as a function of porosity, see Qi Ping et al. Applied Science 2022, 12, 7635; Yun Lin et al. Hindawi Geofluids 2019, 7320536. Although the authors mention the pdf of porosity, e.g. Table 4, they do not present any correlations against porosity or discussion.”

Although the authors have updated the review section, what I really wanted was actual plots of variable against the porosity such as those appearing in the cited references. Please address this point with more analysis and correlations against porosity.

Response 1: Thank you for your valuable feedback on our manuscript. We have addressed your comments as follows:We recognize the importance of presenting and analyzing correlations with porosity. We have provided detailed scatter plots illustrating the relationship between porosity and various parameters, such as pore diameter, pore area, and circularity. These plots were inspired by the methodology employed in the references you cited (Qi Ping et al., Applied Science 2022, 12, 7635; Yun Lin et al., Hindawi Geofluids 2019, 7320536). We have expanded the discussion section to include a comprehensive analysis of these correlations. We have provided interpretations for the observed trends and their implications for the structural changes in the rock samples under different immersion conditions. This detailed discussion, as you requested, addresses the need for a more thorough analysis and correlation with porosity.

We are grateful for your valuable feedback. We look forward to your further comments and suggestions.

Point 2: Figure 3: (D) is simply a repeat of A-C with fits added. Please reduce this to a single figure3D=figure 3.Figure 3: make the axes and table insert fonts bigger (or put the table into the main text) – they are not clearly visible. All figures: increase the axes font size for the same reason as above.

Response 2: Thank you for your suggestion. I have enlarged and standardized the font size in all figures throughout the paper to ensure clarity and consistency for reader accessibility.

Point 3: “About 90% of the references are studies in China published in Chinese journals; I have found these difficult to access, many I could not access at all. It is important that references, particularly those containing data, in an internationally renowned journal such as PONE should be readily accessible to the international community. The authors should make sure that their references are accessible internationally.”

The two references that I suggested were just examples. The authors should find more similar references. Although the authors have referenced more international journals, there are still some that are difficult to access e.g. references 16. Furthermore, some of the web links are spurious – for example, the links in references 38-42 seem to be wrong. I do not have the time to go through all the references – the authors should make sure that all 42 references are readily accessible internationally and are properly linked.

Response 3: I have carefully checked and revised all references to guarantee that they are correctly formatted, hyperlinked, and lead to the correct online sources.

I am immensely grateful to the editors and reviewers for their invaluable advice, which has enabled me to meticulously revise the manuscript, restructure its content, and enhance both the academic rigor and readability of the paper. I extend my sincere thanks once again to the editorial team and reviewers for their diligent efforts. Each of your comments has proven to be immensely valuable and instrumental in refining and improving our work. I remain hopeful that our paper will soon gain full recognition and be published.

Dr. Qingsong Zhuo

Xi'an City, Shaanxi Province, China 

Aug. 5, 2024

---

## [Editor Report · Decision Letter 2]

14 Aug 2024

Study on the corrosion behavior and mechanical response of weakly cemented sandstone in alkaline solutions

PONE-D-24-04355R2

Dear Dr. Zhuo,

We’re pleased to inform you that your manuscript has been judged scientifically suitable for publication and will be formally accepted for publication once it meets all outstanding technical requirements.

Kind regards,

Elena Marrocchino

Academic Editor

PLOS ONE
---

## [Editor Report · Acceptance letter]

23 Aug 2024

PONE-D-24-04355R2 

PLOS ONE

Dear Dr. Zhuo, 

I'm pleased to inform you that your manuscript has been deemed suitable for publication in PLOS ONE. Congratulations! Your manuscript is now being handed over to our production team.

Kind regards, 

on behalf of

Dr. Elena Marrocchino 

Academic Editor

PLOS ONE